# The Effect of Visualization Techniques on Students of Occupational Therapy during the First Visit to the Dissection Room

**DOI:** 10.3390/healthcare10112192

**Published:** 2022-11-01

**Authors:** Juan José Criado-Álvarez, Carmen Romo-Barrientos, Carmen Zabala-Baños, Manuela Martínez-Lorca, Antonio Viñuela, Isabel Ubeda-Bañon, Alicia Flores-Cuadrado, Alberto Martínez-Lorca, Begoña Polonio-López, Alicia Mohedano-Moriano

**Affiliations:** 1Integrated Care Management, Castilla-La Mancha Regional Health Services (SESCAM), 45600 Talavera de la Reina, Spain; 2School of Health Sciences, University of Castilla-La Mancha, 45600 Talavera de la Reina, Spain; 3Department of Medical Sciences, Ciudad Real Medical School, Regional Center for Biomedical Research, University of Castilla-La Mancha, 13071 Ciudad Real, Spain

**Keywords:** anatomy, visualization technique, prosection, cadaver, anxiety, occupation

## Abstract

Background: Part of the basic teaching of human anatomy are prosection sessions with a human corpse, which may generate stress or anxiety among students. The objective of this work was to study how, through the visualization technique (a coping technique), these levels could be reduced before starting prosection classes. Methods: A cross-sectional pilot study was conducted involving first-year students who had never participated in screening sessions. Prior to the visit, occupational therapy students underwent a viewing session (visualization technique). On the day of the visit, before and after the screening session, an anonymous questionnaire was distributed to find out about aspects of the students’ experiences, such as their feelings and perceptions. The State–Trait Anxiety Inventory was used to assess anxiety. Results: The baseline levels of anxiety measured remained stable (from 18.5 to 18.2 points), with no differences being found (*p* > 0.05). The levels of emotional anxiety measured fell from 15.2 to 12.6 points (*p* < 0.05). Before starting the class, there were six students (17.1%) with anxiety criteria, and this figure was doubled at the end of the session (33.3%) (*p* < 0.05). Conclusions: Sessions in a dissection room can cause stressful experiences and change the emotional balances of some students. The results obtained and published here showed no significant differences after the visualization technique. We found that the students believed that the prosection sessions were very useful for teaching anatomy.

## 1. Introduction

Human anatomy is an attraction for most health sciences students, as it will provide their first contact with the human body in their university career [1,2,3,4], although they consider it a difficult subject due to the large amount of information to be assimilated, the memory-based learning, the difficulty in terms of spatial orientation [5], and the anatomical terminology—a problem in other, related subjects, such as histology—and due to the insufficient teaching time [6]. On the other hand, dissection sessions performed on human cadavers are part of a long academic tradition [7], and these sessions are considered a good learning tool for most anatomy professionals [3,7,8] and students of different health science degrees [4,8,9,10,11]. Nevertheless, they can be a stressful experience due to seeing, touching, and feeling anatomical preparations, and can involve various moral issues [12,13,14]. The practical classes carried out in an anatomical dissection room are usually part of the study plans of medicine degrees but not of other degrees related to human sciences, such as occupational therapy, where the organization of these practices is complex for technical reasons [15,16]. Dissections or prosections performed on a human body are some of the first death experiences of future professionals, which can sometimes cause anxiety and stress [1,12,13,16,17]. Reactive feelings and anxiety levels related to the dissection room have been studied on different continents and to varying degrees, and also with the implementation of various coping techniques [1,10,11,12,13,14,15,16,17,18,19,20,21]. Coping techniques enable the management of emotions and behaviors related to anxiety situations, such as in the dissection room, so they must be able to integrate the negative emotions caused by experiences through awareness and understanding of them and the development of effective strategies to deal with them immediately and in the future, because coping can indirectly influence the formation of professional–patient relationships [4]. Situational anxiety reactions (normal responses, given the situation of entering the dissection room) can condition a student’s decision to attend practical sessions and can also affect the performance of the practice since the student must pay attention to the task and their own indicators of anxiety [22], so the implementation of coping techniques is recommended. There have been previous studies on coping techniques to reduce anxiety in the dissection room, such as preparatory audiovisuals [14,16,23], reintegrating humanizing forces [24,25], living testimonies of the so-called “corpse experience” [26,27], and the expression of experiences in the dissection room [4,28]. Currently, the visualization technique has not been valued as a coping technique for the dissection room. This technique is very effective in controlling anxiety, induces relaxation, and is used to help individuals learn to stay calm and to relax in stressful situations. In this sense, visualization exercises to reduce anxiety are easy to learn [29,30,31]. Visualization techniques not only imply suggesting an image or imagining something; they are processes that involve all the senses (sight, smell, hearing, taste, and touch). The practice is typically made up of a series of guided phases of physical relaxation and attention control, beginning with an introductory stage, followed by a process of progressive relaxation (induction), using imagery, deepening techniques (further relaxation), and symptom-specific suggestions to reduce anxiety, stress, pain, etc. [28], where the mind is very open and focused and memory is enhanced [32]. The physiological benefits include the reduction of anxiety and its physiological correlates, stress, control, and pain symptoms. It improves concentration, memory, self-confidence, and positive attitude formation [29,30,33,34,35,36,37].

One of the situations that generates the most stress and anxiety is not knowing what is going to happen. The purpose of visualization is to imagine a more comfortable situation so that when the real situation arrives, it can be faced with more security [38,39,40].

The objective of this study was to apply the visualization technique among first-year occupational therapy students before carrying out their dissection practices in the cadaver room (a stress-generating situation), in order to provide them with control tools that would contribute to better coping with the situation and greater learning or benefit through its practice.

## 2. Materials and Methods

### 2.1. Study Design, Setting, and Participants

A pilot study with a cross-sectional, descriptive epidemiological before–after-type design was conducted during the 2017/2018 academic year with first-year OT undergraduate students enrolled on a course in Anatomy and Human Physiology (annual, 12 ECTS, 120 h, of which 40 are practical classes) at the School of Human Sciences of the University of Castilla-La Mancha (UCLM) in Talavera de la Reina (Toledo, Spain). Practical prosection classes were held in the dissecting room at the Ciudad Real Medical School, which is part of the UCLM. These classes lasted 4 h and were divided into two 2 h sessions (Session 1: “The locomotor system and neuroanatomy”; Session 2: “Splanchnology”).

On the day before the practical session, the visualization technique was carried out in the Occupational Therapy classroom by two psychologists trained in this technique. The visualization technique is based on imagining in a very vivid way a situation in the most real way possible, considering and controlling all variables, so that the person in question perceives themself as capable, safe, and in control of their body and mind [41]. Here, this consisted of a visualization process [35,36,42] to work on reducing anxiety and coping with a practical prosection class with a human corpse in the days before first contact. The visualization process lasted about 15 min and included three time points, focus, deepening, and stabilization, which were presented using symbolic language, metaphors, analogies, and visualizations. The content was related to the physical characteristics of the environment in which the screening session would take place and what the students would feel, experience, see, smell, hear, and touch. After the visualization, the students were given some time (if needed) to get back in touch with reality and leave the room whenever they wanted.

### 2.2. Participant Recruitment Procedure

All participants (*n* = 36) in the pilot study were given an anonymous and invalidated questionnaire, based on the model of Miguel-Perez et al. from 2007 and used by other authors [10,11,12,43], which was designed for the study of the feelings and emotions that they felt about the practical session [11]. To assess their states of anxiety, the State–Trait Anxiety Inventory (STAI) was used. The STAI is a self-administered questionnaire that has been validated in Spanish with Cronbach’s alphas for TA and SA of 0.93 and 0.92, respectively [44], and is conceived as a research instrument to study anxiety in healthy adults. It consists of two scales of 20 items each (state anxiety (SA) and trait anxiety (TA)). TA measures the basic feelings of an individual, while SA evaluates how someone feels when faced with a given stressful situation. The absolute value of the difference in the values obtained between both provides information on whether an event causes anxiety. The dissection session was considered to generate anxiety if the value obtained by the individual exceeded 10 points (STAI-Total > 10) [45,46,47]. These two questionnaires were administered before and after entering the dissection room. The students were informed of the general objectives of the study, and the study was approved by the Clinical Research Ethics Committee of Talavera de la Reina (Toledo) (CEIC File 23/2017). 

### 2.3. Data Management and Analyses

The descriptive statistical analysis used variable scale parameters (single frequencies, measures of central tendencies, and standard deviations). For the analysis of the distribution of variables, Kolmogorov–Smirnov tests were applied to study normal distributions. For the inferential statistical analysis of independent variables, ANOVA tests were used to study the relationship between a normal continuous variable and a nominal one. If the outcome variable was dichotomous, the Student’s *t*-test was used. To study the differences between paired continuous variables, a paired *t*-test was used, and when comparing nominal and dichotomous variables, a chi-squared test was used. A confidence interval of 5% was established. For data analysis, the SPSS statistical package, version 29.0 for Windows (IBM Corp., Armonk, NY, USA), was used. 

## 3. Results

Thirty-six UCLM Occupational Therapy undergraduates (72%) enrolled on the Anatomy and Human Physiology course participated in the study (academic year 2017–2018). Their mean age was 19.9 ± 2.26 years (median: 19 years; range: 18–31 years); 32 (88.9%) were female, with a mean age of 20.0 ± 2.71, and 4 were male (11,1%), with a mean age of 19.7 ± 2.36. 

The results of the STAI questionnaires showed that the basal anxiety (TA) levels remained stable and showed no differences (*p* > 0.05) throughout the session and went from 18.5 to 18.2 points for the 2017/2018 academic year (Table 1). The emotional anxiety levels measured by SA fell from 15.2 to 12.6 points (*p* < 0.05) (Table 1).

Before the dissecting session, there were six students (17.1%) with anxiety criteria (STAI-Total > 10), which doubled at the end of the session to 12 students (33.3%) (*p* < 0.05); that is, the students who previously showed anxiety remained anxious at the end. However, the group’s overall emotional anxiety (SA) lowered from 15.2 to 12.6 points (*p* < 0.05).

The results for the anonymous “ad hoc” questionnaires reflected the students’ subjective perceptions, showing that eighteen students (51.4%) reported having thought about death while participating in the session, but only five students (13.9%) mentioned having felt afraid of losing control. In relation to these objective anxiety figures, we can see that, according to Table 2, before the practical session, one student (2.9%) stated experiencing anxiety, which is a lower figure than that obtained with the STAI. Despite these emotions, 85.7% (*n* = 30) of the students were curious about the class and 88.6% felt emotionally prepared to enter the dissecting room. Feelings also changed during the class (Table 3); the number of students not feeling nervous rose from 15 (42.9%, before) to 31 (81.6%, after) (*p* < 0.05), and the number feeling scared increased from 22 (62.9%) to 33 (91.7%) (*p* < 0.05). The feeling of relaxation also grew after the dissection session, the number of students who felt relaxed increasing from 18 (51.4%) to 30 (83.3%) (*p* < 0.05). A total of 71.4% of the participants reported dreading the smell of the dissecting room, while 45.7% said they were afraid of seeing the dead person’s face.

Finally, 33 students (91.7%) recommended this prosection session for future year groups, and 33.3% (*n*: 12) and 66.7% (*n*: 24) felt “Satisfied” or “Very satisfied”, respectively. The mean score for the experience (measured from 0 to 10) was 8.9 ± 1.14 points (median: 9; range: 4–10).

## 4. Discussion

Anatomy is a basic subject taught in most health science degree courses. Dissections or prosections are excellent ways of learning human anatomy. Being able to interact directly with a body in the dissection room allows for better understanding and ways of spatially relating to human structures [10,15,48,49,50]. Such interactions allow students to promote self-reflection and help engage the cognitive and affective skills required for professional practice [48,51].

Entering a dissecting room can be a big step that challenges some students’ emotional equilibria [14,19,46,51,52]; for some students, it may even be their first contact with a corpse [12,13,18]. However, our data show that the students considered the experience to be satisfactory or very satisfactory and they recommended it for future years (100%) [9,53,54,55]. In our study, students’ emotional anxiety levels decreased from 15.2 points to 12.6 points (*p* < 0.05), which results were similar to those reported in other published works [10,14,17,21]. The fact that students’ emotional anxiety levels fell could be the result of their finishing their first practice session, and, after a rest period, exchanging opinions with their classmates, “socializing with death” [2,4,47]. Some significant differences emerged before and after in both academic years (*p* < 0.05). These differences have been found by some authors [14,17,52]. None of the study participants admitted feeling emotionally unprepared to enter the dissection room for the first time. Only 11.4% felt indifferent compared to 88.6% who admitted feeling emotionally prepared, unlike another study that reported that 64% of the participants did not feel emotionally prepared [56].

It is also worth noting that 85.7% of our students felt curiosity about entering the dissecting room, which is a similar result to those obtained in other studies [14,21,57,58]. Before and after, there were slight increases in positive feelings, such as feelings of calm, happiness, and comfort, without these being statistically significant, which data correlate with those of other studies for health science degrees, such as medicine and physiotherapy. The students believed that the “smell of the dissecting room” was unpleasant (71.4%), similar to findings reported elsewhere, with between 60% and 90% reporting this sensation [9,19,59]. However, it is striking that dreading the smell of the dissecting room reached higher values than those indicated by Romo-Barrientos [21], whose study conditions were like those of our own. Finally, 45.7% of the students indicated that they dreaded seeing the face of the corpse, which is again similar to other previously reported results [12,60], the face being related to the personality of the cadaver and students exhibiting greater emotional reactions to dissecting the face [61,62]. In addition, Moxham [63] suggests that students initially easily personalize the corpse and do not see it as a mere educational “tool”. This experience causes students to ask questions, such as existential, emotional, philosophical, and religious ones [64,65]. These reactions in medical students are channeled through a process of psychological accommodation by having several sessions in the dissection room; on the other hand, students on the Occupational Therapy degree course attend only once or twice and do not have time for this process of accommodation to occur. On the method of psychological accommodation, see refs. [66,67].

Previous studies have shown that preparation seminars (using audiovisual means) can help students reduce their stress (SA) levels to greater extents than those who follow more conventional formats [14,23,47,68,69]. Another proposed alternative is to play background music in the dissecting room [70], to humanize the process [24,66,71], or to provide a prior description of the most significant aspects that students will experience (smell or touch), which could help lower their anxiety levels [72]. 

The visualization technique favors positive attitudes toward coping and problem-solving. It entails engaging the imagination and emotional involvement, and favors a state of relaxation in a way that reduces stress [29,30]. 

The aim of the visualization technique, used as a tool, is to achieve control of the mind, reduce anxiety or stress and tension, and relieve pain. Anxiety and stress generate negative perceptions or thoughts and images [73]. Positive visualizations, on the other hand, cause the brain to generate substances to feel better, such as serotonin [31].

No studies have contemplated the visualization technique as a coping technique to help students face the challenge of entering a dissecting room, despite its being recommended to solve AS problems. The closest thing has been the use of seminars to prepare students for what is going to be found in the room using audiovisual media [47]. In our case, we opted for the visualization technique as a coping strategy to lower anxiety levels with a possible positive impact, other studies having shown decreases in the anxiety generated by competition between young players [74] and, furthermore, that visualization technique sessions led to observations of academic performance improvement and reduced anxiety about evaluation tests [75,76].

In the study by Romo-Barrientos [20], the SA levels obtained with the visualization technique (coping therapy) were similar to those of a previous study [21], where SA levels fell from 14.6 to 10.0 (without coping therapy). 

It is worth noting that the students who had been prepared with the visualization technique obtained higher TA and SA levels in both of the phases before and after prosection compared to those in the previous study [21] who went without coping therapy. One possible explanation is that the visualization technique may require training, since not all people have the capacity for imagination or inventiveness. These training periods will be longer for people with more problems, in whom all the senses must be stimulated to create a visualization that is as real as possible [40]. In this academic-year analysis, the authors focused on seeing anxiety and did not analyze whether there were students who had problems imagining a system to reduce their stress or anxiety, and perhaps this meant that we did not generate significant data.

One limitation of the present study is its small sample size (36 students, 72% of those enrolled for OT), and there may also be limitations regarding the external validity of the results. The sample represents the total reference population; therefore, the results can be extrapolated to populations of occupational therapy students similar to ours.

Another limitation is that, as this was the second time such practical sessions had been carried out, the students could have been informed by the students from the previous year. It was also the first academic year in which this coping technique was implemented, and perhaps more training or a larger number of sessions might be required. External factors could also have influenced the results, for example, the environment and the presence of experienced professionals before the prosection session, the self-preparation for visualization, and previous warnings about preparing for the first experience of prosection to generate confidence among the students. The final limitation is that no previous results are available. 

## 5. Conclusions

Practical sessions held in dissecting rooms promote positive learning in students [77] and are considered a basic part of the practice of teaching human anatomy [51,78]. Although a low percentage of students were emotionally affected by coming into contact with a corpse, most considered that participating in this activity was valuable for their training, and their degree of satisfaction was very high. This session is not an encounter with a patient and could prove to be an unforgettable experience that favorably affects their future professional work as occupational therapists. It is thus necessary to start including such valuable sessions in the anatomy courses offered as part of degrees in OT. Moreover, teachers/educators should bear in mind such student reactions to be able to detect them early on and efficiently deal with them using a variety of coping strategies.

Surprisingly, the visualization technique is a technique recommended to deal with problems related to states of mind such as anxiety [79,80,81], and, although the absolute numbers showed a decrease in state anxiety, no differences due to the induction technique were effective, such that we must continue to search for effective strategies for our students and future professionals. In fact, future researchers could study other useful strategies for reducing anxiety symptoms, such as emotional ventilation and creating spaces where students can express their emotions and feelings and share experiences with classmates and teachers to lay the foundations for the humanization of professional practices. There are also other therapies, such as mindfulness, meditation, and yoga, as well as creative techniques (writing, painting, role playing), music, etc. [56], which could help to reduce anxiety in our students in the dissection room.

## Figures and Tables

**Table 1 healthcare-10-02192-t001:** States of anxiety of the students in the moments before and after the practice.

	Before	After	
	Mean ± Standard Deviation	Mean ± Standard Deviation	Statistical significance
TA	18.5 ± 7.21	18.2 ± 7.28	*p* > 0.05
SA	15.2 ± 6.01	12.6 ± 8.16	*p* < 0.05
STAI SUM	33.7 ± 11.73	30.8 ± 12.66	*p* < 0.05
STAI Total	5.3 ± 4.83	9.0 ± 5.77	*p* < 0.05

TA: Trait Anxiety; SA: Anxiety State; STAI SUM: Sum of the values of AR and AE.

**Table 2 healthcare-10-02192-t002:** Student’s thoughts before the practical session.

		Yes		No	
		*n*	%	*n*	%
Thinking about dissection evokes	Anxiety	1	2.9	34	97.1
	Displeasure	1	2.9	34	97.1
	Curiosity	30	85.7	5	14.3
	Uncertainty	16	45.7	19	54.3
	Fear	3	8.6	32	91.4
Which is the most unpleasant experience in the dissecting room?	Seeing the cadaver’s face	16	45.7	19	54.3
	The smell of the dissecting room	25	71.4	10	28.6
	Touching the cadaver	3	8.6	32	91.4

In none of these cases were there statistically significant gender differences regarding the students’ thoughts (*p* > 0.05).

**Table 3 healthcare-10-02192-t003:** Students’ feelings during the prosection practice. * The data is statistically significant.

	Before						After						
	Yes		Indifferent		No		Yes		Indifferent		No		Statistical Significance
	*n*	%	*n*	%	*n*	%	*n*	%	*n*	%	*n*	%	
I feel calm	25	71.4	10	28.6	0	0	35	97.2	1	2.8	0	0	*p* > 0.05
I feel confident	30	85.7	5	14.3	0	0	33	91.7	3	8.3	0	0	*p* > 0.05
I feel nervous	5	14.3	14	42,9	15	42.9	1	2.8	4	11.1	31	86.1	*p* < 0.05 *
I feel scared	2	5.7	11	31.4	22	62.9	0	0	3	8.3	33	91.7	*p* < 0.05 *
I feel happy	27	77.1	8	22.9	0	0	29	80.6	6	16.7	1	2.8	*p* > 0.05
I feel comfortable	30	85.7	5	14.3	0	0	34	94.4	1	2.8	1	2.8	*p* > 0.05
I feel relaxed	18	51.4	16	45.7	1	2.9	30	83.3	5	13.9	1	2.8	*p* < 0.05 *
I feel worried	2	5.7	11	31.4	22	62.9	4	11.1	3	8.3	29	80.6	*p* > 0.05
Do you feel emotionally prepared for entering the dissecting room?	31	88.6	4	11.4	0	0	30	83.3	4	11.1	2	5.6	*p* > 0.05

## Data Availability

Not applicable.

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
