# Peer review of "The Effect of Visualization Techniques on Students of Occupational Therapy during the First Visit to the Dissection Room"

_healthcare, 2022, doi:10.3390/healthcare10112192_

Round 1

Reviewer 1 Report (New Reviewer)

Thank you for the possibility to review the manuscript titled: “The effect of coping techniques on students of Occupational Therapy during the first visit to the dissection room.” The manuscript provides important insights for teaching anatomy in medical universities. There if only one minor point, wchich requires further analysis:

-The fact that students are usually more afraid to see the face of the cadaver needs more analysis in the discussion section. This is an interesting phenomenon from a psychological point of view and is important to describe in more details in discussion.

Please take into account the recommendation in the spirit of improving the quality of the submission.

Author Response

A paragraph has been incorporated into the discussion on the indicated topic.

Reviewer 2 Report (New Reviewer)

1.          In the section of the introduction, I suggest the author could provide more information about the learning difficulties or disadvantages of human anatomy for health sciences students.

2.          In addition, I suggest the author should provide more literature analysis or theoretical explanation about coping techniques and their advantages or usefulness.

3.          In the section of the method, I suggest the author should provide more information about the theoretical backgrounds or literature analysis of the survey tools.

4.          In the section of data management and analyses, I suggest the author should provide more information about the validity and reliability of the research data.

5.          In the section of results, some statistical analysis did not meet the significant level. I suggest the authors should explain this issue with theoretical discussions.

6.          In the section of the conclusion, I suggest the author should provide some practical strategies for this issue.

Author Response

Point 1

Learning difficulties have been added to the discussion 

Point 2

Added the perks of visualization coping techniques

Point 3

In material and methods in point 2.2, the information from the questionnaires that were supplied to the students has been added. The STAI is a questionnaire validated in several languages and widely used for the study of anxiety. There is a high number of works carried out with this questionnaire. 

Point 4

The questionnaire itself has been used in various articles to assess feelings and emotions before entering the dissection room and published in journals such as Annals of Anatomy, Science Education, European Annals of Otorhinolaryngology, and Head and Neck diseases. BMC Medical Education and the data have been analyzed by a statistician.

Point 5  

A comment has been added to the discussion about the indicated point.

Point 6

Some practical strategies for this topic have been added to the conclusions

Reviewer 3 Report (New Reviewer)

The manuscript reports the effects of coping techniques among students of occupational therapy upon first encounter on human dissection.

My comments are as follows:

1. In the abstract, there are no indications of Occupational Therapy student group and coping techniques. These keywords must be mentioned.

2. Coping mechanism has been well-elucidated in these articles:

(i) https://anatomypubs.onlinelibrary.wiley.com/doi/10.1002/ase.1332

(ii) https://bmcmededuc.biomedcentral.com/articles/10.1186/s12909-015-0355-9

(iii) https://www.sciencedirect.com/science/article/abs/pii/S094096021830133X?via%3Dihub

Therefore the gap of study that leads to current research has to be elaborated in the Introductory section

3.  Visualisation technique is the only coping mechanism investigated in this study. It is advisable to mention this technique in the main title rather than "coping techniques"

4. There is a need to elaborate visualisation technique in the Introduction

4. The study was conducted in academic year of 2017/2018. What is the reason of delaying the publication of these findings? Are these observation still valid after 4-5 years?

5. Any questionnaires involved? - This has to be include in the Methodology

6. 2.3 Statistical analysis

Please clarify/revise this section as to include percentage differences between the measured variables, the use of ANOVA, independent-sample t-test and chi-squared test (if any) and the n number

7. To include a Table on Distribution of OT students by year of study, age and gender

8. Limitations and future perspectives - this section shall be placed in the Discussion. Elaborate the issue of small sample size in this study.

9. Coping techniques must not be a separate heading in the Discussion. If visualisation technique is the only coping technique involved, please focus only on the former.

Author Response

Point 1

In the abstract been added what was requested

Point 2

Coping techniques in the management room have been added to the introduction, adding the appropriate bibliography

Point 3

The title has been modified due to the recommendation it indicated. Removed Coping Technique for Visualization Technique

Point 4

Added visualization technique in the intro.

Point 4 bis

Once the 2017-2018 academic year has ended, occupational therapy students have not been able to do internships in the dissection room due to Covid-19. In the current academic year, he has not resumed 100% face-to-face teaching.

These data are valid because it is the only study of this type.

Point 5

In Matter and method, the two questionnaires used have been explained. One is to assess feelings and emotions in the dissection room which has been used in different articles and has been cited. The STAI questionnaire is validated and described in the methodology.

Point 6

Statistical analysis has been developed in greater detail.

Point 7

No demographic table has been added as all data has been added in the text. If you insist on putting the table it would be like the one we indicated.

Male

Female

Students of Terapia ocupacional

4

32

Age

19.7±2.36

20.0±2.71

The study only refers to the 2017-2018 academic year. An academic year. 

Point 8

The limitations section has been removed and the text has been incorporated into the discussion and the problem of sample size has been explained. One limitation of the present study is its small sample size (36 students, 72% of those enrolled in OT), although it may produce limitations regarding the external validity of the results. The sample represents the total reference population, so there are no validity problems since the results can be extrapolated to populations of occupational therapy students similar to ours.

Point 9

Your suggestion has been added

Round 2

Reviewer 2 Report (New Reviewer)

The author made more revisions to this manuscript based on my concerns. I suggest this manuscript should be accepted in the present edition.

Reviewer 3 Report (New Reviewer)

-

This manuscript is a resubmission of an earlier submission. The following is a list of the peer review reports and author responses from that submission.

Round 1

Reviewer 1 Report

 In the present study, the authors try to reduce the anxiety generated in 36 students of 1st year of OT during a 4-hour practice of prosections, for which, before the practice, they are given a 15-minute session using the visualization technique.

Most of the content of the work has been previously published. See:

Criado-Álvarez JJ, González González J, Romo Barrientos C, Ubeda-Bañon I, Saiz-Sanchez D, Flores-Cuadrado A, Albertos-Marco JC, Martinez-Marcos A, Mohedano-Moriano A. Learning from human cadaveric prosections: Examining anxiety in speech therapy students. Anat Sci Educ. 2017 Sep;10(5):487-494. doi: 10.1002/ase.1699. Epub 2017 May 4. PMID: 28472535.

Romo-Barrientos C, Criado-Álvarez JJ, Gil-Ruiz MT, González-González J, Rodríguez-Hernández M, Corregidor-Sánchez AI, Ubeda-Bañon I, Flores-Cuadrado A, Mohedano-Moriano A, Polonio-López B. Anatomical prosection practices in the Occupational Therapy degree. Student anxiety levels and academic effectiveness. Ann Anat. 2019 Jan;221:135-140. doi: 10.1016/j.aanat.2018.10.003. Epub 2018 Oct 10. PMID: 30315912.

Romo Barrientos C, José Criado-Álvarez J, González-González J, Ubeda-Bañon I, Saiz-Sanchez D, Flores-Cuadrado A, Luis Martín-Conty J, Viñuela A, Martinez-Marcos A, Mohedano-Moriano A. Anxiety among Medical Students when Faced with the Practice of Anatomical Dissection. Anat Sci Educ. 2019 May;12(3):300-309. doi: 10.1002/ase.1835. Epub 2018 Oct 30. PMID: 30378293.

Romo-Barrientos C, Criado-Álvarez JJ, González-González J, Ubeda-Bañon I, Flores-Cuadrado A, Saiz-Sánchez D, Viñuela A, Martin-Conty JL, Simón T, Martinez-Marcos A, Mohedano-Moriano A. Anxiety levels among health sciences students during their first visit to the dissection room. BMC Med Educ. 2020 Apr 9;20(1):109. doi: 10.1186/s12909-020-02027-2. PMID: 32272926; PMCID: PMC7146885.

Romo-Barrientos C, Criado-Álvarez JJ, Martínez-Lorca A, Viñuela A, Martin-Conty JL, Saiz-Sanchez D, Flores-Cuadrado A, Ubeda-Bañon I, Rodriguez-Martín B, Martinez-Marcos A, Mohedano-Moriano A. Anxiety among nursing students during their first human prosection. Nurse Educ Today. 2020 Feb;85:104269. doi: 10.1016/j.nedt.2019.104269. Epub 2019 Nov 15. PMID: 31760350.or

As an example, the data from the control group used in this study have already been published in Romo-Barrientos et al. (2019) and Romo-Barrientos et al. (2020).

The only novelty of the work is the use of the visualization technique to reduce the level of anxiety, but it is a "poor" intervention to be able to validate it as an alternative to those already proposed.

Although the authors, with the exception of the works by Arráez-Aybar et al., is the first intervention to reduce anxiety levels in students of OT, I would recommend that they review the bibliography given that there are numerous studies on the emotional reactions of medical students to the dissecting room, as well as proposals to reduce stress and anxiety levels (e.g. Boeckers et al., 2010; Bourguet et al., 1997; Charlton, 1994; Dickinson et al, 1997; Dosani & Neuberger, 2016; Gonzalez-Pinilla et al., 2020; Horne et al., 1990; Houwink et al., 2004; Hull, 1991; Lazarus et al., 2017; Mark et al., 1997; Wisenden et al., 2018, etc.).

Reviewer 2 Report

This manuscript builds on previous works of the Authors in which they studied anxiety levels in several health sciences students. In particular, this paper aims to evaluate the reduction of anxiety levels by applying a coping technique (visualization) in occupational therapy students attending a prosection course.

The are several issues that prevent the publication of this paper in the present form.

The major criticism concerns the evidence that setting, participants, results and conclusions appear superimposable to previously published works by the same authors. In addition, It is not clearly demonstated if novelty (application of the coping technique) here introduced actually produced significant results. Therefore, the manuscript seems lacking of originality.

The following  issues should also be considered:

The abstract is not fully consistent with the text; the application of coping strategies in methods section is missing; several sentences of the template are present and should be deleted (line 25, line 31).

The overall structure of the manuscript is difficult to follow, the phrases and the periods are often too long. 

The terms prosection and dissection are frequently used interchangeably in the manuscript but the research design employ only the prosection techniques. In this regard, the AA should consider that even if prosection and dissection are often strongly recommended  for the students of the degree course in Medicine nevertheless,  it has been widely shown that in three-year degree courses the educational objectives can also be achieved using virtual dissection techniques, avoiding anxiety and other problems. In fact, as suggested by several and recent studies, digital technologies are efficient in learning anatomy, lowering anxiety.  This modern teaching approach should be discussed in the context of the paper, in order to complete the information and provide to the readers different points of view and recent evidence on this topic.  

The paper needs an extensive re-edition mainly addressed to demonstrating the originality and consistency of new results and possibly by collecting data from a larger sample of students to better evaluating the efficiency of coping techniques and reinforcing the  original data.